# A Source of Systematic Errors in the Determination of Critical Micelle Concentration and Micellization Enthalpy by Graphical Methods in Isothermal Titration Calorimetry

**DOI:** 10.3390/e23020236

**Published:** 2021-02-18

**Authors:** Mónica Corea, Rogelio Jiménez-Juárez, Gabriela Martínez-Mejía, María de Jesús Martínez-Ortiz, José Manuel del Río

**Affiliations:** 1Programa de Nanociencias y Micro-Nanotecnologías, Instituto Politécnico Nacional, Escuela Superior de Ingeniería e Industrias Extractivas, UPALM, Edificio Z-5 PB, San Pedro Zacatenco, Del. Gustavo A. Madero, C.P. 07738 Ciudad de Mexico, Mexico; mcoreat@yahoo.com.mx (M.C.); mjmartinez@ipn.mx (M.d.J.M.-O.); 2Departamento de Química Orgánica, Instituto Politécnico Nacional, Escuela Nacional de Ciencias Biológicas, Prolongación de Cárpio y Plan de Ayala s/n Miguel Hidalgo, C.P. 11340 Ciudad de Mexico, Mexico; rjimenezg@ipn.mx (R.J.-J.); gmmejia@outlook.es (G.M.-M.); 3Laboratorio de Investigación en Polímeros y Nanomateriales, Instituto Politécnico Nacional, Escuela Superior de Ingeniería Química e Industrias Extractivas, UPALM, Edificio Z-5 PB, San Pedro Zacatenco, Del. Gustavo A. Madero, C.P. 07738 Ciudad de Mexico, Mexico

**Keywords:** isothermal titration calorimetry, critical micelle concentration, micellization enthalpy, micelle formation

## Abstract

Isothermal titration calorimetry is frequently employed to determine the critical micelle concentration and the micellization enthalpy of surfactants in terms of geometrical characteristics of the titration curves. Previously we have shown theoretically that even for an infinitesimal injection, the heat per titrant mol depends on the stock solution concentration. In this work, we explore experimentally the influence of the stock solution concentration on the geometrical characteristics of the titration curve and its effect in determining the critical micelle concentration and the micellization enthalpy of surfactants. The systematic study of this phenomenology involves a great number of measurements at different temperatures with several repetitions carried out using a robotic calorimeter. As surfactant hexadecyltrimethylamonium bromide was used. The magnitude and shape of the heat titration depend on the stock solution concentration. As a consequence, the inflexion-point, break-point, and step-height decrease until a limiting value. A qualitative analysis suggests that the limiting value depends only on substance. This work shows that graphical methods could not be suitable for the calculation of the critical micelle concentration and micellization enthalpy because the magnitude and shape of the titration curve depend on the stock solution concentration. Micellar properties should be calculated by the application of theoretical models as in the ligand-binding studies.

## 1. Introduction

Isothermal titration calorimetry is an experimental technique which measures the heat involved in a titration process [1,2]. Basically, an isothermal titration calorimeter consists of a syringe which contains a titrant and a cell (surrounded by calorimetric detectors) which contains the titrated liquid. A quasi-isothermal condition [3] is achieved by compensating electrically the heat flow absorbed or released from each titration process. The first isothermal titration calorimeters [1,2] were developed to study chemical reactions. Other applications such as macromolecular binding [1,2], liquid mixtures [1,4,5] and micelle formation [1,2] were also developed. In this work, we are interested in the application of isothermal titration calorimetry to the study of micelle formation thermodynamics.

The experiment of micelle formation, such as we know (see Figure 1), was originally proposed by Olofsson [6,7,8] who built the first titration curves for micelle formation and provided a phenomenological interpretation which is still employed nowadays. This interpretation is based upon the idea that changes in the titration curve reflect changes in the aggregation state. Recently some authors have given experimental evidence about this relation by fluorescence [9], small-angle neutron scattering [10], small-angle X-ray scattering [10], scanning transmission electron microscopy [10] as well as Monte Carlo simulation [10]. Olofsson also proposed a method calculating the enthalpy of micelle formation (Δ*_mic_**h*) and a method to obtain a value of the critical micelle concentration (*CMC*).

Van Os and coworkers [13] using an automatic isothermal titration calorimeter, proposed other methods for calculating Δ*_mic_**h* and the *CMC*. Both methods were derived from other experimental techniques. The graphical method for obtaining Δ*_mic_**h* was taken from that developed by the group of Desnoyers [14,15] for Picker calorimeters [16,17] and is based upon the following Equation (see Appendix B for details):(1)$$L_{2}^{a}-L_{2}^{b}=\;\Delta_{mic}h+(m-m_{CMC})\left(\frac{\partial \Delta_{mic}h}{\partial m}\right)\;$$
where *L^a^*_2_ and *L^b^*_2_ are the linear extrapolations from the micellar region and from the monomer region, respectively, and m is the concentration on the molality scale. In order to get Δ*_mic_**h* by Equation (1), the extrapolated values *L^a^*_2_ and *L^b^*_2_ must be evaluated at the *CMC*. Because a value of the *CMC* is required, van Os and coworkers developed a criterion to determine it which was adapted from techniques such as electrical conductivity [18], sound speed [19], or surface tension [20]. In this method, the *CMC* is calculated from the intercept of the linear extrapolations from the monomer and micellar regions in the cumulated heat of titration. This method has been used by many other authors [21,22,23]. In addition, Equation (1) established the “standard method” used hereafter in the bibliography: the enthalpy of micelle formation is obtained by the difference between the linear extrapolations from the monomer and micellar regions evaluated at the *CMC*.

In 1995, Paula and coworkers [24] proposed another method for calculating the *CMC* by isothermal titration calorimetry. These authors observed that the minimum (or maximum) of the first derivative of the titrant heat per titrant mol (Q/n^s^) with respect to the concentration is in the transition region between monomer and micellar regions, and for this reason, they proposed this point as *CMC*.

In 1998, Kresheck [25] proposed using an empirical fit function to obtain the *CMC* as the inflexion point of the titrant heat per titrant mol. Other improvements of this method were proposed by Király and Dekány [26] and by Textor and Keller [27].

Isothermal titration calorimetry is a recent and widely used technique which provides a relatively quick determination of the *CMC* and Δ*_mic_**h* with a good degree of accuracy [28]. For this reason, it is interesting to study in detail Equation (1) which is the thermodynamic foundation of the “standard method.” Appendix B shows that *L*_2_ in Equation (1) is calculated by a partial derivative of the Euler Equation for the relative enthalpy *L* of the system without applying the corresponding Gibbs–Duhem Equation to cancel null terms. The reorganization of these null terms in Equation (1) gives the artificial term (*CMC*-m)(∂Δ*_mic_**h*/∂*m*) which is also null. For this reason, the correct form of Equation (1) is:(2)$$L_{2}^{a}-L_{2}^{b}=\;\Delta_{mic}h$$

Equation (2) is not suitable for graphical evaluation because if the monomer and micellar region are not flat, the difference *L^a^*_2_—*L^b^*_2_ will depend on the concentration and Δ*_mic_**h* will be undetermined. In addition, it is not possible to directly apply Equation (1) to isothermal titration calorimeters because the titration heat per titrant mole is not always the partial relative enthalpy of the solute [1,2]. This is only true when the titrant is a pure liquid compound [1,2,4,5].

On the other hand, an argument in favor of the standard method has been proposed by some authors [25,29]. In these works, the *CMC* was obtained in terms of inflexion points and Δ*_mic_**h* was obtained by means of the difference between the extrapolations at the *CMC*. To probe the thermodynamic consistency, they used the van’t Hoff Equation to predict the *CMC* from Δ*H*_m_ and Δ*Cp*_m_ obtaining agreement within the experimental error. This result suggests that certain geometrical characteristics of the titration heat curves could be considered as reasonable approximations of micellar properties. Our problem now is to determine whether the geometrical characteristics of the titration curves depend only on the substance or also on the experimental design of the titration process.

From a theoretical approach, it is possible to see qualitatively if the stock solution concentration can affect the titration curve. By notation, in the work we will use the subindex C16-TAB to refer to the total complex solute (surfactant). This complex solute is composed by surfactant in monomer state and surfactant in micellar estate. In the case of a 2-component system, the solvent will be component 1 and the solute will be component 2. Considering a 2-component system without reaction or aggregation, the titration heat per titrant mole in an infinitesimal titration is [1,2]:(3)$$\frac{dQ}{dn_{2}^{s}}=\frac{\rho_{2}^{s}-c_{2}^{s}}{c_{2}^{s}}h_{1;2}(c_{2})+h_{2;1}(c_{2})-\frac{1}{c_{2}^{s}}h_{v}(c_{2}^{s})$$
where components 1 and 2 are respectively the solvent and solute, *c*_2_ and *c^s^*_2_ are respectively the concentration of solute in the titration cell and in the stock solution, *h*_1;2_ and *h*_2;1_ are respectively the molar enthalpy of 1 and 2 and *h_v_*(*c^s^*_2_) is the enthalpy per unit volume of the stock solution. The stock solution concentration in Equation (3) could affect the titration curve by a vertical shift by means of the term (1/*c^s^*_2_)*h_v_*(*c^s^*_2_). In addition, it could affect the shape of the titration curve. The relative titration heat is defined as:(4)$$\frac{dQ_{L}}{dn_{2}^{s}}=\frac{dQ}{dn_{2}^{s}}-{\left(\frac{dQ}{dn_{2}^{s}}\right)}^{0}$$
where the extrapolated value (*dQ*/*dn^s^*_2_)^0^ is defined as:(5)$${\left(\frac{dQ}{dn^{s}}\right)}^{0}\equiv \lim_{c_{2}\to 0}\frac{dQ}{dn^{s}}$$

Substituting (3) and (5) in Equation (4) the relative titration heat can be written as:(6)$$\frac{dQ_{L}}{dn_{2}^{s}}=\frac{\rho_{2}^{s}-c_{2}^{s}}{c_{2}^{s}}l_{1;2}(c_{2})+l_{2;1}(c_{2})$$
where *l*_1;2_ and *l*_2;1_ are the relative partial enthalpies [30] of the components 1 and 2 respectively, which depend only on the substance. From Equation (6), the shape of the relative titration heat curve could be affected by the stock solution concentration by means of the term (**ρ^s^**_2_ − *c^s^*_2_)/*c^s^*_2_. That is, even for a simple model of a 2-component system without reaction or aggregation, the value of the stock solution concentration can change the magnitude and shape of the heat titration curve.

As it was said, the titration curve shape is related to structural changes in micellar aggregates [9,10]. On the other hand, different theoretical approaches of the titration heat, in terms of the apparent molar heat [31,32], or in terms of infinitesimal titrations [1,2], indicate the dependence of the titration heat curve on stock solution concentration. To clarify how the stock solution concentration affects the geometrical properties of the titration curves, we have systematically scanned the proposed phenomenology using a robotic calorimeter. With this instrument we can change the stock solution concentration within an appropriate experimental range, repeating each titration curve four times at different temperatures to determine the effect on Δ*_mic_**Cp*. All titration curves have been measured in the same range of concentration (independently of the stock solution concentration) and all curves had a minimum amount of points before (and after) *CMC*. This study provides experimental evidence about the effect of the stock solution concentration on the break-point, the inflexion point, and the height of the step of the heat titration curves.

## 2. Material and Methods

Because the number of runs is large, we selected an inexpensive surfactant with a low *CMC* (1 mM): hexadecyltrimethylammonium bromide (C16-TAB; Fluka > 99%). The water was twice distilled and then deionized using a MicroPure ST Barnstead purification system from ThermoScientific.

The calorimeter was an Automated MicroCal PEAQ-ITC from Malvern Panalytical (Northampton, MA, USA). The sample cell is coin-shaped with a volume of 200 microliters and the titration syringe has a volume of 40 microliters. The level noise in the thermogram is around 0.63 nW and the temperature stability at 25°C is around ±0.00012 °C. This robotic instrument is equipped with a sample tray containing four sample plates each with 48 sample wells. One robotic arm controls and cleans the titration syringe and another two robotic arms clean the sample cell and take the samples from the sample tray to use them in the cell and in the syringe. This instrument can work continuously without supervision following a preset work program.

To check the effect of the stock solution on the titration curve six stock solution concentrations were prepared from 5 mM to 40 mM in flasks of 100 mL (see Table 1).

The measurement of a titration curve was considered as a run. For low stock solution concentrations, the titration volume is relatively large and so we carried out a small number of titrations with a completely full syringe. In these cases, the run had to be split into several sub-runs. Between sub-runs, the robotic system subtracted an amount of liquid from the cell and refilled the syringe. Sub-run thermograms were merged using the Malvern Panalytical software MicroCal Concat ITC (version 1.0). Table 1 shows the experimental parameters of the runs. All titration curves were measured from 0 to 3 mM. For this reason, when the stock solution concentration changed, all experimental parameters must also vary (see Table 1) to avoid undesirable side effects and to take advantage of the instrument robotization. For example, if the titration volume and the number of titrations are constant while the stock solution varies, the concentration interval of the titration curves will differ, producing systematic errors in the data processing. In the same way, if the concentration interval and the titration volume are held constant, the number of titrations will change dramatically, again generating systematic errors in the data processing. As a criterion, a reasonable variation in the number of titrations per run was allowed while ensuring the monomer region contained at least 10 points.

To take advantage of the robotization, the number of repetitions for each stock solution concentration and temperature was kept at four to provide an additional source of precision. A water-water test was interspersed every two repetitions to check the correct performance of the equipment. The heat registered in these water-water runs is produced by the friction of the fluid in the syringe needle [2]. For each temperature, the total number of sub-runs, including the water-water test, was 72, and the total number of sub-runs in this work was 216. The time duration of a water-water test including the cleaning of the cell and the syringe is around 2 h, while the duration of a sub-run depends on the number and duration of the titrations. The total time to complete a work program was around one and half weeks, which required the robotic calorimeter to work continuously.

The data processing was carried out with methods implemented by algorithms. This automatization of the calculus also avoids the bias introduced by the subjectivity, experience and/or appreciation of the operator in the application of the methods. In this work, the break-point of the monomer region and the inflexion point of the titration curve will be calculated as measurements of the *CMC*. As a measure of the micellization enthalpy, we will calculate the high step of the titration curve at the *CMC*. All methods were programed in Excel spreadsheets.

In the break-point method (Figure 2A), the titration curve was divided into three regions. Regions 1 and 3 were defined in terms of empirical fit functions, and region 2 was considered as a transition region. In the first region, the titration heat per mole of titrant was constant in all measured titration curves. The following criterion was employed to define this region. Initially, the group of the first six points was taken. A new point was included in the group if it lay within the interval [x_A_ − 3σ, x_A_ + 3σ] where x_A_ and σ are the average and standard deviation of the new group and so on. This method stops when the new point does not fall in the interval [x_A_ − 3σ, x_A_ + 3σ] indicating the end of the monomer region. The points of region 3 were fitted to the empirical function:(7)$$f(x)=\frac{A}{x-B}+C$$
where *A*, *B* and *C* are parameters which were obtained minimizing the function: (8)$$\chi^{2}=\sum_{i}{\left[y_{i}-f(x_{i})\right]}^{2}$$

This calculation was carried out using the SOLVER of Excel with the following parametrization. The method in the minimization was the GRG Nonlinear, the precision restriction was 1 × 10^−250^ and the scheme of derivatives was central. The micellar region was defined in the same way as the monomer region. After the fit of the function (7) to the six latest points, the following point is included in the region if it is within the interval of three standard deviations around the value *f*(*x*). The standard deviation [38] was estimated according to the following Equation:(9)$$\sigma^{2}\approx \frac{1}{m-n}\sum_{i=1}^{n}{\left[y_{i}-f(x_{i})\right]}^{2}$$
where *m* is the number of parameters of the fit function (in this case 3), and *n* is the number of points. The following points were included following the same criterion. The *CMC* was calculated by the intercept of the constant value of the first region and the function *f*(*x*) of the third region (see Figure 2A). Comparing Figure 1 and Figure 2A, the proposed method yields values of the break-point bigger than that of Figure 1 using two straight lines. Other differences are that the method implemented by two straight lines depends on the number and position of the points of the transition region and on the subjective appreciation of the operator to define the transition region. The proposed method does not depend on these characteristics and as consequence is numerically more stable and easier to program.

The inflexion point method was implemented in the following way. Figure 2B shows the first derivative of the titration curve with respect to the concentration of C16-TAB. The shape of this experimental first derivative was described in this work as an asymmetric Gaussian function:(10)$$g(x)=A\exp\left\{-{\left(\frac{x\;-\;CMC}{\sigma_{0}\;+\;\sigma_{1}x\;+\;\sigma_{2}x^{2}\;+\;\sigma_{3}x^{3}}\right)}^{2}\right\}$$
where σ_0_, σ_1_, σ_2_, σ_3_, and *CMC* are fit parameters. In this way, the experimental points of the titration curve will be fitted to the empirical function:(11)$$f(x)=\int_{0}^{x}g(x^{\prime })dx^{\prime }+\sigma_{4}$$
where σ_4_ is the other fit parameter. All fit parameters of Equations (10) and (11) were calculated by minimizing the function χ^2^ of Equation (8) using the SOLVER of Excel and the integral of Equation (11) was calculated numerically using the rectangle rule. The parameter *CMC* in Equation (10) plays the role of a minimum implying that it takes the role of an inflexion point in Equation (11).

The high step of the titration curves, as a measure of Δ*_mic_**h*, was calculated using the method proposed by van Os [13] evaluated at the break-point and at the inflexion point (Figure 3). The linear regions were defined in a similar criterion of the three standard deviations around the fit as in the break-point method. 

Another thermodynamic property calculated from the titration curves was the extrapolated value (*Q*/*n^s^*_C16-TAB_)^0^ obtained by Equation (5) using the linear fit of the first region (Figure 3).

## 3. Result and Discussion

Figure 4 shows the averaged titration curves at 25 °C (Figure 4A), 35 °C (Figure 4B), and 45 °C (Figure 4C). At all temperatures, the height of the monomer and micellar regions increases with the stock solution concentration. Because this increment is bigger in the monomer region than in the micellar region, it is clear that the step height (a measure of Δ*_mic_**h*) is affected by the stock solution concentration. In addition to this, it is clear in Figure 4 that the length of the monomer region (a measure of the *CMC*) decreases with the stock solution concentration. 

It is usual to measure the titration curve at different temperatures to study the effect of the temperature on the *CMC* and on Δ*_mic_**h* [24,25,26,29,37]. The heat capacity of micellization is obtained by plots of the enthalpy of micellization against the temperature. These studies were carried out preparing only one stock solution. Figure 5 shows two examples of the effect of temperature on the titration curve for two different stock solution concentrations indicating clearly the dependence of this type of study on the stock solution concentration. The arbitrariness is evident because the step height at 45 °C for the stock solution concentration of 5.0 mM is similar to the step height at 35 °C for the stock solution concentration of 25.0 mM. 

Figure 6 shows the effect of the stock solution concentration on the extrapolated value defined in Equation (5). The extrapolated value depends linearly on the temperature in the temperature interval studied (see Figure 6A) while the slope of the linear fits depends on the stock solution concentration. It is interesting to observe that the extrapolated value for 5.0 mM at 35 °C is practically the same as the value for 25.0 mM at 25 °C. Figure 6B shows the extrapolated value as function of the stock solution concentration at several temperatures. In all cases, the extrapolated value increases with the stock solution concentration up to a limiting value. 

Figure 7 shows the break-points and inflexion points calculated from the titration curves of Figure 4. The results from these two methods are similar. Figure 7A shows the break-points and inflexion points as a function of the temperature for two different stock solution concentrations. The arbitrariness of these values as function of the temperature is evident (see Figure 7A) because the values for 5.0 mM at 25 °C are similar to the values for 25.0 mM at 35 °C. Figure 7B shows the break-points and inflexion points with respect to the stock solution concentration at different temperatures. In all cases, the values decrease with the stock solution concentration until a limiting value. This result is consistent with the length reduction of the first plateau in the titration curves in Figure 4 when the stock solution concentration increases. Although it is usual that the authors do not report the exact concentration of the stock solution, it is interesting to analyze the inflexion point results of this work in the context of the literature data. Figure 8A shows the inflexion point data for 5.0, 25.0, and 30.0 mM and data of *CMC* from the bibliography (Appendix A). All bibliography data were obtained by isothermal titration calorimetry using an inflexion-point method of a break-point method [23,28,33,34,35,36,37]. Except for one point that clearly falls out of the general trend, all the literature data are between the values measured with the stock solution concentrations of 5.0 mM and 30.0 mM, indicating that the effect of the stock solution concentration on the inflexion point explains the scattering of the *CMC* literature data for the C16-TAB. 

Figure 9 shows the step height of the titration curves calculated by the two both methods exposed in the Material and Methods Section. No appreciable differences between methods were found. Figure 9A shows the step height as a function of the temperature for two stock solution concentrations. In both cases, the step height depends linearly on the temperature as in the case of the extrapolated value (Figure 6). Similar results, where the enthalpy of micelle formation is linear, or close to a linear behavior, was found by other authors [24,25]. From Figure 9A, the slope of the fit depends on the stock solution concentration indicating that the heat capacity calculated from this method will depend on the stock solution concentration. In addition to this, the arbitrariness of the micellization enthalpy calculated from the step height is evident because the value at 35 °C for a stock solution concentration of 25.0 mM is similar to that of 45 °C for 5.0 mM. Figure 9B shows the step height as a function of the stock solution concentration at several temperatures. In all cases, the step height decreases until a limiting value.

Figure 8B compares the step height values obtained in this work and literature values calculated by isothermal titration calorimetry using similar methods [8,23,28,33,34,35,36,37] (Appendix A). Except for a few points between 25 and 30 °C, all data are between the values measured with the stock solution concentrations 5.0 and 25.0 mM indicating that the dependency of the step height with the stock solution concentration can explain the scattering of the Δ*_mic_**h* literature data for the C16-TAB.

Usually, the micellization heat capacity [24,25] is calculated by the slope in plots of the step height of titration curves as a function of the temperature. Figure 10 shows this amount dΔ_Step_/dT (obtained by the two methods discussed in the Material and Method Section) as function of the stock solution concentration. Figure 10 shows that this amount decreases until a limit value when the stock solution concentration in the syringe increases.

Until now, we have seen that the stock solution concentration affects the magnitude and geometrical characteristics (break-point, inflexion-point, and step height) of the heat titration curve, but we have not studied the thermodynamic cause of this effect. It is possible to study qualitatively this behavior using the relative titration heat and considering the micellar system as a 2-component system in the monomer region and in the micellar region (without considering the details of the micelle formation).

Figure 11 shows the relative heat titration curve at several stock solution concentrations calculated using Equation (6) and values of the extrapolated value shown in Figure 6. In the monomer region, before the *CMC*, the system can be considered as composed of the solvent (component 1) and surfactant monomer in solution (component 2). Because in this region the relative heat capacity is zero independently of the stock solution concentration, by Equation (6) this region can be described by the following Equation:(12)$$0=\frac{\rho^{s}-c_{2}^{s}}{c_{2}^{s}}l_{1;2}(c_{2})+l_{2;1}(c_{2})$$

Differentiating with respect to *c*_2_ in (12), considering the Gibbs–Duhem Equation for the relative enthalpy in a binary system:(13)$$\left(\rho_{2}-c_{2}\right)\frac{dl_{1;2}}{dc_{2}}+c_{2}\frac{dl_{2;1}}{dc_{2}}=0$$
where ρ_2_ is the molar density of the system at the concentration *c*_2_ and bearing in mind that *l*_1;2_(0) = *l*_2;1_(0) = 0, it is obtained that both relative partial molar enthalpies of solvent and solute must be identically zero below *CMC*. This means that the system of C16-TAB in water behaves as an ideal solution in the monomer region independently of the stock solution concentration.

After *CMC* micelle formation is a consequence of solute-solute interaction. Appendix C shows that for a diluted solution with solute-solute interactions, the partial relative enthalpy of solute and solvent are not null and that the contribution of the relative partial enthalpy of solute is bigger than that of the relative partial enthalpy of solvent. For this reason, both non-zero contributions must be considered in Equation (6). In addition, considering that:(14)$$\frac{\rho_{2}^{s}-c_{2}^{s}}{c_{2}^{s}}=\frac{1}{x_{2}^{s}}-1$$
and substituting (14) in Equation (6) and taking the limit of *x^s^*_2_ approaching to 1:(15)$$\lim_{x_{2}^{s}\to 1}\frac{dQ_{L}}{dn_{2}^{s}}=l_{2;1}$$

Figure 11 shows that *dQ_L_*/*dn_s_* decreases with the stock solution concentration until limiting curve. This behavior can be explained by means of Equation (15) because when the stock solution concentration increases enough, the relative heat of titration approaches to the relative partial enthalpy of solute. This result can explain the limiting values obtained for the inflexion point, break-point, and step height. For low stock solution concentrations, the contribution of the term (ρ*^s^*_2_—*c^s^*_2_)/ *c^s^*_2_ goes from 1850 (for 30.0 mM) to 11,000 (for 5.0 mM) and because the relative partial enthalpy is non-zero, the term (ρ*^s^*_2_—*c^s^*_2_)/ *c^s^*_2_ × *l*_1;2_ can affect the value of relative heat of titration in Equation (6) explaining the general dependence of the inflexion point, break-point, and step height with the stock solution concentration. It is possible to understand this behavior evaluating all contributions with the following numerical example. Figure 12 shows the behavior of the relative titration heat at 25°C evaluated at one arbitrary concentration in the cell (for example 2 mM), as a function of the stock solution concentration in the syringe. The curve was calculated from the fit of the relative titration heat against the molar fraction of the stock solution concentration using the fit function:(16)$$f(x)=\left(\frac{1}{x}-1\right)A+B$$
where *A* and *B* are constant. The obtained values of the constants are *A* = 2.6 × 10^−4^ ± 0.1 × 10^−4^ kJ/mol and *B* = −10.52 ± 0.08 kJ/mol and they can be interpreted, respectively, as the relative molar enthalpies of the solvent and the solute evaluated at the concentration of 2 mM. As it was stated by the theorem of Appendix C, the absolute value of the relative partial molar enthalpy of solute is five orders of magnitude bigger than the relative partial enthalpy of solvent. The solvent contribution to the relative titration heat is the term (1/*x^s^*_2_ − 1) × *l*_1;2_ and at the stock solution concentration of 5 mM this term takes the value 2.88 kJ/mol while at 30 mM it decreases until 0.48 kJ/mol. Figure 12 shows clearly that the relative titration heat is the balance between the solvent contribution which decreases with the stock solution concentration and the solute contribution which is independent of the stock solution concentration. A possible explanation of the dependence of the titration heat with the stock solution concentration could be in terms of a possible change in the morphology of the aggregates in the stock solution. In this work, it was shown that the behavior exposed in Figure 12 can be explained by means of non-ideal solvent effects because the term (1/*x^s^*_2_ − 1) × *l*_1;2_ increases very fast when the stock solution concentration decreases. In addition to this, the asymptotic behavior can be easily understood in the following and intuitive terms. As it was said the titration heat is exactly the relative molar enthalpy when the titrant is a pure compound. When the stock solution concentration increases, the titrant approaches the pure compound of solute. For this reason, the titrant heat approaches the relative enthalpy of solute. 

About the concept of *CMC*, Charles Tanford [39] wrote around fifty years ago: “The concept of a “critical micelle concentration” for the formation of micelles from free amphiphile is inexact but convenient. The use of this concept is probably a major cause of confusion in the thermodynamic analysis of micelle-forming system… It is customary to define a single concentration within this transition zone as a critical micelle concentration, generally abbreviated as “*CMC*.” This is usually done by empirical graphical procedures. These graphical methods were basically proposed by Williams [40] and by Phillips [41] around seventy years ago. In Williams’s approach [40,42] the critical micelle concentration is obtained by the intercept of the extrapolations from both monomer region and micellar region using data from an experimental technique as a function of the surfactant concentration. In this work, we have used this approach implemented in the method of break-point. In the Phillips approach, the *CMC* is defined as the concentration of surfactant in the inflexion point of a particular property measured experimentally:(17)$${\left(\frac{d^{3}\phi }{dc_{T}{}^{3}}\right)}_{c\;=\;CMC}=0$$
where ϕ is a property measured experimentally and *c_T_* is the total concentration of surfactant. In this work, we have used this approach implemented in the method of “inflexion-point”. The *CMC*, such as was defined by Williams and Phillips (and discussed by Tanford), is intimately linked to a particular experimental technique. In this work, we have shown experimental evidence about the dependence of the titration curve with the stock solution concentration. If the *CMC* is defined as a geometrical characteristic of the titration curve obtained from a graphical method (break-point or inflexion-point), a concept of *CMC* which depends on the stock solution concentration is obtained. The concept of micellization enthalpy is different because it is defined as:(18)$$\Delta_{mic}h=h_{mic}^{o}-h_{mon}^{o}$$
where *h^o^_mic_* and *h^o^_mon_* are, respectively, the standard enthalpies of the surfactant in micellar state and in monomeric state. By Equation (18) the micellization enthalpy only depends on substance, indicating that any systematic error is associated with the graphical method used. A way to avoid the dependence of *CMC* and Δ*_mic_**h* with the stock solution concentration is by the employment of theoretical models. This discussion resembles, somehow, the way used in ligand binding studies where the binding equilibrium constant and the binding enthalpy are calculated by different theoretical models. In the simplest case of the interaction of ligand and a macromolecule with a stoichiometry 1:1, studied by means of the Wiseman Isotherm [43], the characteristics of the titration curve depend dramatically on Brandt’s parameter [44,45] defined as:(19)$$c=c_{M}K_{Bind}$$
where *c_M_* is the concentration of macromolecule and *K_Bind_* is the binding equilibrium constant. Graphical methods are not used in this case because the shape and magnitude of the titration curve depends on the experimental design expressed in terms of Brandt´s parameter. In addition to this, the value of this parameter affects the precision and reliability of the values of binding enthalpy and binding equilibrium constant obtained by fit of theoretical models and some authors have proposed different ranges of c in order to get the optimum value of this experimental parameter [45,46,47].

As it was said, the titration curve approaches the relative partial molar enthalpy of solute when the stock solution concentration increases. To use high stock solution concentrations could be a way to avoid systematic errors in the determinations of *CMC* and Δ*_mic_**h*. The problem of this way is that titration volumes would be very small and other systematic errors will appear due to the lack of control in the automatic titration system to give an exact volume. From an operational point of view many times it is not possible to carry out a study similar to that of this work in order to determine the better value of the stock solution concentration. Based upon the fact that in general the shape and magnitude of the titration curve depend on the experimental design, the solution of this problem is to use the theoretical model (as in the case of ligand binding). This is the most recommended procedure to determine the critical micelle concentration and the micellization enthalpy by isothermal titration calorimetry, in special for shorter alkyl chains for which the micellization transition is not so steep (cooperative). For this reason, theoretical models including the stock solution concentration effect [32] should be used to obtain the *CMC* and Δ*_mic_**h* because if this effect is not considered [48], systematic errors could be present in the results.

## 4. Conclusions

In this work, we have studied the influence of the titration process on the determination of the micellar properties using a robotic calorimeter.

It was found that the magnitude and shape of the heat titration curve depend on the stock solution concentration. As consequence geometric characteristics as inflexion point, break-point, and step height depend on the stock solution concentration. For high stock solution concentrations in 2-component systems, the relative heat titration curve approaches to the relative partial enthalpy of solute. This fact can explain the fact that inflexion point, break-point, and step height approach limiting values when the stock solution concentration increases.

As a practical rule, it was proposed that there is a balance between the stock solution concentration and the injection volume. The stock solution concentration should be as big as possible to decrease the effect of non-ideality of the solvent while the injection volume should be as big as possible to improve the control in the injection volume.

In this work, it was found that the calculation of *CMC* and Δ*_mic_**h* by graphical methods can include systematic errors because the dependence of the titration heat on the stock solution concentration in the syringe. This effect is produced by the non-ideal behavior of the solvent after the *CMC*.

Theoretical models on micelle formation should describe the exposed phenomenology. In this way, they could be used in the future to obtain the critical micelle concentration and the micellization enthalpy from experimental titration curves without systematic errors.

## Figures and Tables

**Figure 1 entropy-23-00236-f001:**
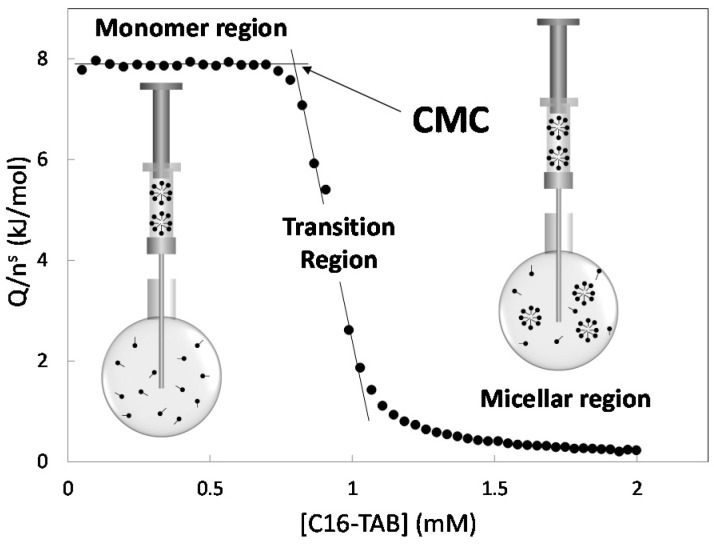
Titration curve at 25 °C obtained in this work using hexadecyltrimethylammonium bromide (C16-TAB) as surfactant. In this type of curve, the titration heat per unit titrant mol is represented against the titrant concentration in the cell. The stock solution concentration in the syringe was 5.0 mM. In this case the critical micelle concentration is around 1 mM. A criterion to obtain the critical micelle concentration (*CMC*) is in terms of the point where the monomer region breaks down [6,7,11,12].

**Figure 2 entropy-23-00236-f002:**
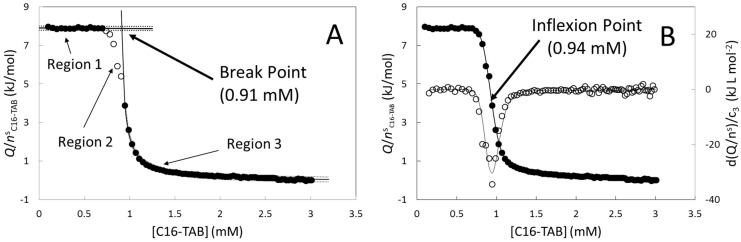
(**A**) break-point method and (**B**) inflexion point method employed to determine the *CMC*. The titration curve of both (**A**) and (**B**) is the same as in Figure 1. Filled circles in (**A**) represent the points included in the monomer and micellar regions. In (**B**) open circles represent the first derivative calculated using a centered finite differences scheme.

**Figure 3 entropy-23-00236-f003:**
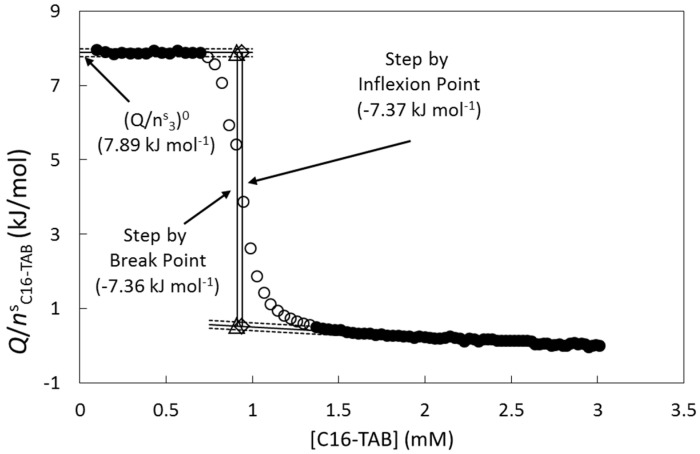
Graphical methods used to calculate the step height of the titration curve. The filled circles are the points included in the linear fits; points not included are open circles. The titration curve was the same as in Figure 1.

**Figure 4 entropy-23-00236-f004:**
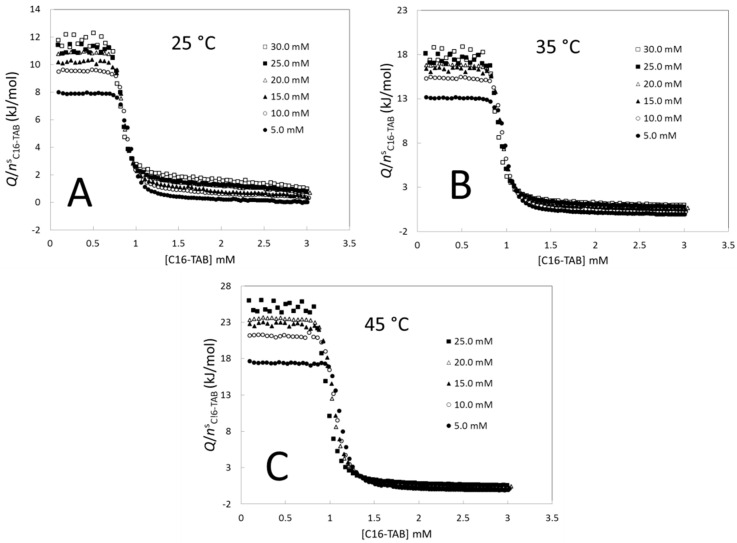
Averaged titration curves of C16-TAB at different stock solution concentrations and temperatures.

**Figure 5 entropy-23-00236-f005:**
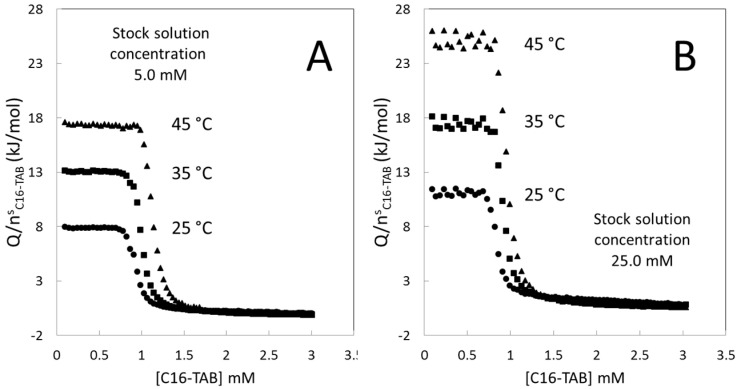
Heat titration curves of C16-TAB at different temperatures for two stock solution concentrations using data of Figure 4. (**A**) the stock solution concentration is 5.0 mM and (**B**) the stock solution concentration is 25.0 mM. This Figure shows how it would be the results of two different authors using two different stock solution concentrations in the same study of micellization of C16-TAB.

**Figure 6 entropy-23-00236-f006:**
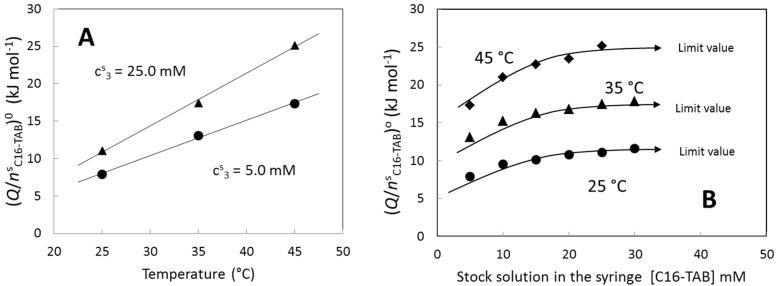
(**A**) Extrapolated value (see Equation (5)) of C16-TAB as function of the temperature for two stock solution concentrations. (**B**) Extrapolated value of C16-TAB as function of the stock solution concentration at several temperatures.

**Figure 7 entropy-23-00236-f007:**
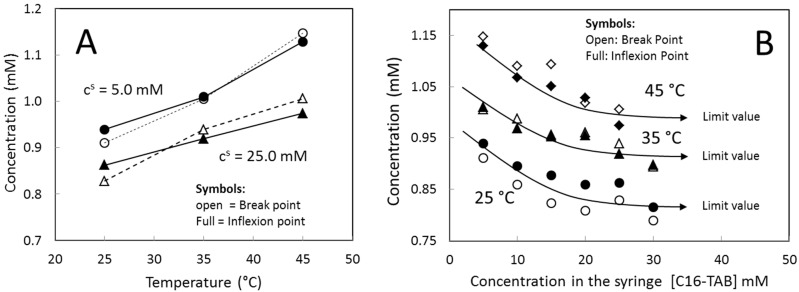
(**A**) Inflexion-point (full symbols) and break-point (open symbols) as function of the temperature for two stock solution concentrations. (**B**) Inflexion-point (full symbols) and break-point (open symbols) as function of the stock solution concentration at several temperatures.

**Figure 8 entropy-23-00236-f008:**
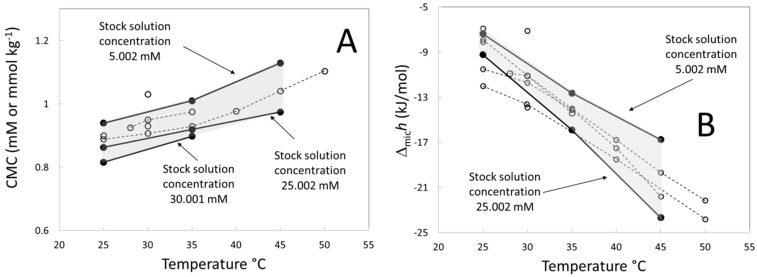
Comparison of literature data and data obtained in this work for C16-TAB. (**A**) Inflexion-points (full circles) and *CMC* values from the literature (open circles). (**B**) Step height measured at the inflexion point (full circles) and enthalpies of micellization obtained from the literature (open circles).

**Figure 9 entropy-23-00236-f009:**
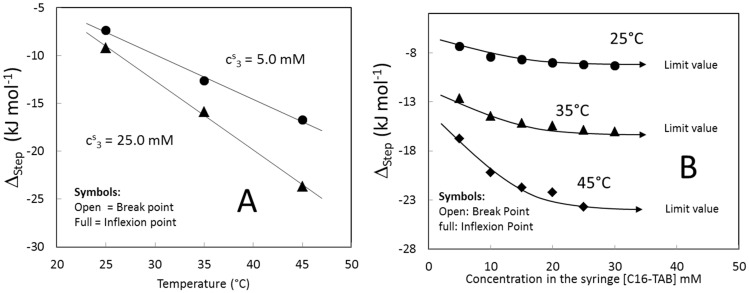
(**A**) Step height of C16-TAB measured at the inflexion point (full symbols) and measured at the break-point (open symbols) as function of the temperature at two stock solution concentrations. (**B**) Step height of C16-TAB measured at the inflexion point (full symbols) and measured at the break-point (open symbols) as function of the stock solution concentration for several stock solution concentrations.

**Figure 10 entropy-23-00236-f010:**
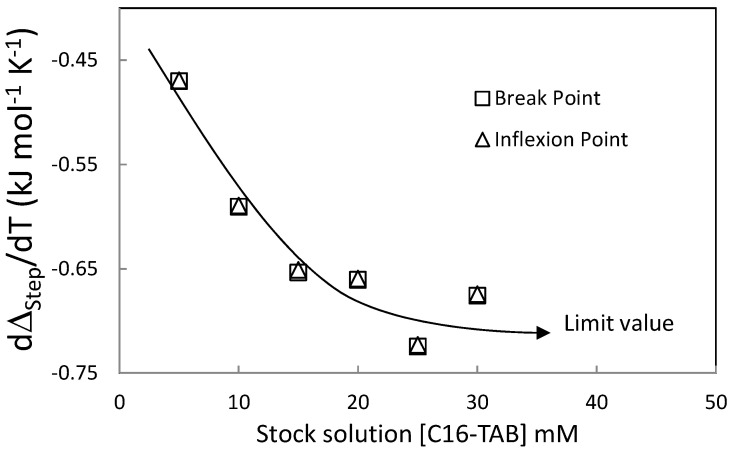
Slope of the step height with respect to the temperature (dΔstep/dT) as function of the stock solution concentration of C16-TAB.

**Figure 11 entropy-23-00236-f011:**
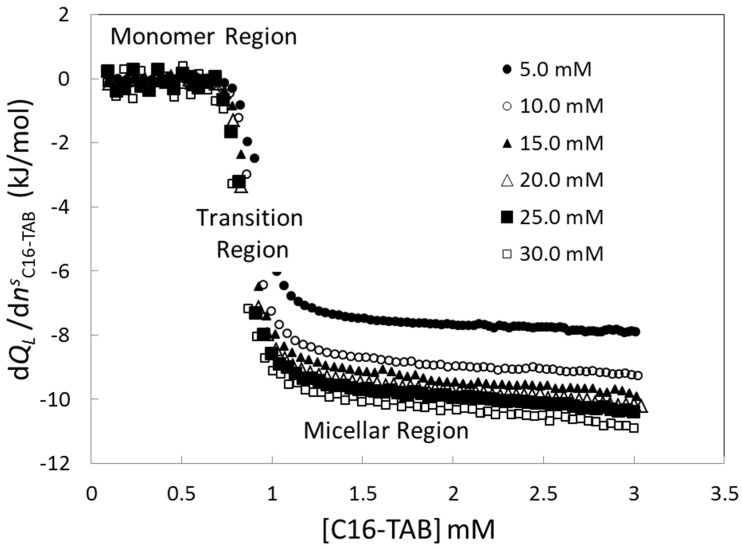
Relative titration heat of C16-TAB at 25 °C at several stock solution concentrations.

**Figure 12 entropy-23-00236-f012:**
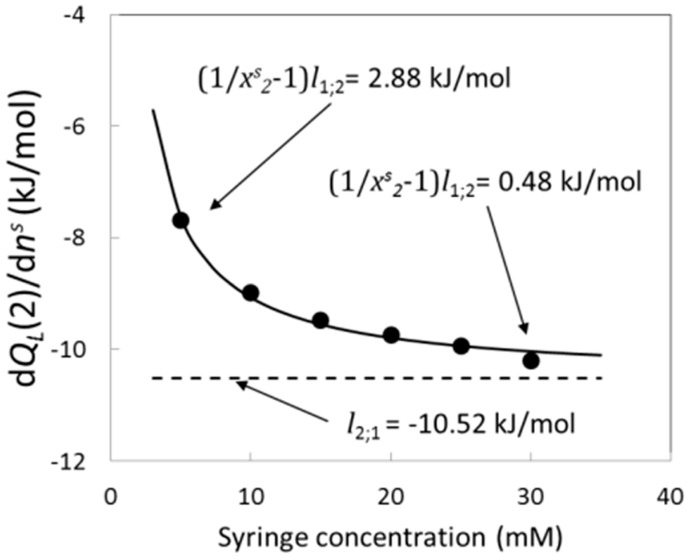
Relative titration heat of C16-TAB at 25 °C evaluated at the concentration of 2 mM as function of the stock solution concentration in the syringe. The value (1/*x^s^*_2_ − 1) *l*_1;2_ is the solvent contribution to the relative titration heat. Full circles represent the relative titration heats and the curve is the fit calculated by Equation (16).

**Table 1 entropy-23-00236-t001:** Parametrization of runs with different stock solution concentrations. The *CMC* of C16-TAB is about 1 mM [23,33,34,35,36,37] and all titration curves were measured in the interval of concentration from 0 to 3 mM of C16-TAB.

Stock Solution Number	Stock Solution Concentration (mM)	Titration Volume (μL)	Injection Duration (s)	Number of Sub-Runs	Number of Titrations Per Subrun
1	5.0	2.00	3.00	5	19
2	10.0	1.00	2.00	2	39
3	15.0	0.67	1.34	2	39
4	20.0	0.50	1.00	1	66
5	25.0	0.37	0.74	1	70
6	30.0	0.31	0.62	1	68

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
