# Peer review of "A Source of Systematic Errors in the Determination of Critical Micelle Concentration and Micellization Enthalpy by Graphical Methods in Isothermal Titration Calorimetry"

_entropy, 2021, doi:10.3390/e23020236_

Round 1

Reviewer 1 Report

This manuscript reports an investigation on how the experimental protocol of an ITC experiment may influence the graphical determination of surfactant cmc values. This is the kind of example that people in the field usually refer to but, to the best of my knowledge, it has not been systematically reported with real data as in this report. Hence, I think this is a useful information, especially for beginners with this technique, what becomes more relevant now that ITC is becoming widespread as one of the most suitable techniques for thermodynamic studies on surfactant micellization.

Overall, the measurements seem to have been properly conducted, their analysis is correctly conducted and I have only minor comments regarding the manuscript organization, as follows:

  • This discussion resembles, somehow, the introduction of the c constant (or parameter) as a product of concentration and K for binding studies using ITC (see, for instance, Keller et al Anal Biochem, 2011 - https://doi.org/10.1016/j.ab.2011.07.027). This could be mentioned here.
  • One should remember that values reported in Tables 2 and 3 (cmc and Delta H values for CTAB micellization) are significantly dependent on the surfactant purity and, hence, even if the same technique were used (let´s assume conductimetry), different values are expected. So, I am not sure how much this adds to the main topic of this manuscript. I would rather stick to the data presented before, which is not dependent on chemicals, equipment or operators, but only on the experimental protocol. Maybe this tabulation may be presented as Supplementary Material.
  • The enthalpy values at the beginning of the titration experiment are different because they reflect different enthalpy values for the surfactant solutions in the syringe (the starting point for this experiment, considering that the final state is the same for all curves, as defined by the concentration at the x-axis). Hence, these values could be used as information related to dilution enthalpy within the concentration range used in the syringe. This is not always realized and I think it may be stressed here. In fact, for literature experiments reported in Tables 2 and 3, it may even be used to estimate what was the initial concentration used, if not reported.
  • Similarly, not all ITC users realize that the values of the enthalpy of micellization derived from the ITC plot should be corrected for the fraction of non-micellized surfactant in the syringe (typically, as high as 5-10%), therefore producing wrong values. This is discussed, among others, in ref. 13 of this Ms.
  • I think there are too many cited references. I believe some were used for the purpose of gathering literature data in Tables 2 and 3, discussed above, but are not directly related to the main purpose of this Ms.
  • Finally, in most the recent reports, ITC curves are analyzed by fitting the data to equations that produce directly cmc and Delta H values. This is in fact, the most recommended procedure, especially for surfactants with shorter alkyl chains, for which the micellization transition is not so steep (or cooperative). This should be more clearly commented in this Ms, and I think this should be the major conclusion of this work, after showing that the shape of the ITC curves depends on experimental protocol. I strongly suggest that the authors consider strengthening this point for the benefit of less experienced users.

Author Response

Reviewer 1-anwers

References were checked and the following paragraphs were included:

Around old line 270: All bibliography data were obtained by isothermal titration calorimetry using an inflexion-point method of a break-point method [23,28,33,34,35-,36,37].

Arround old line 290: Panel B of Figure 8 compares the step height values obtained in this work and literature values calculated by isothermal titration calorimetry using similar methods [23,28,33,34,35,36,37,39] (Table 3 in Supplementary Material).

The following sentence was included arround the old line 96: “By notation, in the work we will use the subindex C16-TAB to refer the total complex solute (surfactant). This complex solute is composed by surfactant in monomer state and surfactant in micellar estate. In the case of a 2-component system, the solvent will be the component 1 and the solute will be component 2.”

You can find the new paragraphs in red in the new version of the manuscript. The anwers of your comments are listed bellow.

This manuscript reports an investigation on how the experimental protocol of an ITC experiment may influence the graphical determination of surfactant cmc values. This is the kind of example that people in the field usually refer to but, to the best of my knowledge, it has not been systematically reported with real data as in this report. Hence, I think this is a useful information, especially for beginners with this technique, what becomes more relevant now that ITC is becoming widespread as one of the most suitable techniques for thermodynamic studies on surfactant micellization.

Overall, the measurements seem to have been properly conducted, their analysis is correctly conducted and I have only minor comments regarding the manuscript organization, as follows:

  • This discussion resembles, somehow, the introduction of the c constant (or parameter) as a product of concentration and K for binding studies using ITC (see, for instance, Keller et al Anal Biochem, 2011 - https://doi.org/10.1016/j.ab.2011.07.027). This could be mentioned here.

The following paragraph was included in the line 394 page 27 of the manuscript:

This discussion resembles, somehow, to the way used in ligand binding studies where the binding equilibrium constant and the binding enthalpy are calculated by different theoretical models. In the simplest case of the interaction of ligand and a macromolecule with a stoichiometry 1:1, studied by means of the Wiseman Isotherm [54], the characteristics of the titration curve depends dramatically on the Brandt´s parameter [54, 55] defined as:

where cM is the concentration of macromolecule and KBind is the binding equilibrium constant. Graphical methods are not used in this case because the shape and magnitude of the titration curve depends dramatically on the experimental design expressed in terms of the Brandt´s parameter. In addition to this, the value of this parameter affects the precision and reliability of the values of binding enthalpy and binding equilibrium constant obtained by fit of theoretical models and some authors have proposed different ranges of c in order of get the optimum value of this experimental parameter [55-56].

  • One should remember that values reported in Tables 2 and 3 (cmc and Delta H values for CTAB micellization) are significantly dependent on the surfactant purity and, hence, even if the same technique were used (let´s assume conductimetry), different values are expected. So, I am not sure how much this adds to the main topic of this manuscript. I would rather stick to the data presented before, which is not dependent on chemicals, equipment or operators, but only on the experimental protocol. Maybe this tabulation may be presented as Supplementary Material.

Tables 2 and 3 where removed from the manuscript and located in the supplementary Material.

  • The enthalpy values at the beginning of the titration experiment are different because they reflect different enthalpy values for the surfactant solutions in the syringe (the starting point for this experiment, considering that the final state is the same for all curves, as defined by the concentration at the x-axis). Hence, these values could be used as information related to dilution enthalpy within the concentration range used in the syringe. This is not always realized and I think it may be stressed here. In fact, for literature experiments reported in Tables 2 and 3, it may even be used to estimate what was the initial concentration used, if not reported.

Figure 6B shows that the extrapolated value depends on the stock solution concentration. But the plot of the extrapolated value against the stock solution concentration cannot be considered as a good calibration curve to estimate the stock solution concentration. From Figure 6B, up to 20 mM the uncertainty in the determination this method cannot determine the stock solution concentration with precision. This is due to the fact that the extrapolated value increases until a limiting value. For this reason when the value of the extrapolated value approaches the limiting value the proposed method is not operative.

  • Similarly, not all ITC users realize that the values of the enthalpy of micellization derived from the ITC plot should be corrected for the fraction of non-micellized surfactant in the syringe (typically, as high as 5-10%), therefore producing wrong values. This is discussed, among others, in ref. 13 of this Ms.

I do not use this correction because I have never understand this correction. In my opinion this concept is dark and not rigorously defined.

  • I think there are too many cited references. I believe some were used for the purpose of gathering literature data in Tables 2 and 3, discussed above, but are not directly related to the main purpose of this Ms.

11 references where removed from the manuscript

  • Finally, in most the recent reports, ITC curves are analyzed by fitting the data to equations that produce directly cmc and Delta H values. This is in fact, the most recommended procedure, especially for surfactants with shorter alkyl chains, for which the micellization transition is not so steep (or cooperative). This should be more clearly commented in this Ms, and I think this should be the major conclusion of this work, after showing that the shape of the ITC curves depends on experimental protocol. I strongly suggest that the authors consider strengthening this point for the benefit of less experienced users.

The following paragraph was included in the line 397 of page 17:

As it was said, the titration curve approaches to the relative partial molar enthalpy of solute when the stock solution concentration increases. To use high stock solution concentrations could be a way to avoid systematic errors in the determinations of CMC and Dmich. The problem is that titration volumes would be very small and other systematic errors will appear due to the lack of control in the automatic titration system. From an operational point of view many times it is not possible to carry out a study similar to that of this work in order to determine the better value of the stock solution concentration. Based upon the fact that in general the shape and magnitude of the titration curve depend on the experimental design, the solution of this problem is to use theoretical model (as in the case of ligand binding). This is the most recommended procedure to determine the critical micelle concentration and the micellization enthalpy by isothermal titration calorimetry, in special for shorter alkyl chains for which the micellization transition is not so steep (cooperative). For this reason, theoretical models including the stock solution concentration effect [31] should be used to obtain the CMC and Dmich because if this effect is not considered [48], systematic errors could be present in the results.

Reviewer 2 Report

I appreciate the idea and work made by the authors. In my opinion, the article is certainly useful to the scientific community and deserves publication. The paper is well written and well structured. The title clearly describes the
contents of the paper. The abstract provides a concise and complete summary and the reference list is well chosen. The presentation is clear and the language is fluent and precise. 

I recommend this paper for publication. However, I do have a couple of comments that should be considered prior to publication:

1.line 49 sentence: "She also proposed a method calculating the enthalpy of 49
micelle formation ...." Not clear "who".

2. I recommend deleting some obvious information about the ITC from the introduction.

3. Line 102, Line 226, Line 341: One style in symbols is recommended

4. I would also mention that also time after which the prepared solution is used affects titration curve shapes. What I mean by that is that cmc values, as well as titration curve shapes, are different for experiments made after one hour one day, and one week. 

In my opinion, the cmc is not a point is rather a region.  

Author Response

Reviewer 2-report

References were checked and the following paragraphs were included:

Around old line 270: All bibliography data were obtained by isothermal titration calorimetry using an inflexion-point method of a break-point method [23,28,33,34,35-,36,37].

Arround old line 290: Panel B of Figure 8 compares the step height values obtained in this work and literature values calculated by isothermal titration calorimetry using similar methods [23,28,33,34,35,36,37,39] (Table 3 in Supplementary Material).

The following sentence was included arround the old line 96: “By notation, in the work we will use the subindex C16-TAB to refer the total complex solute (surfactant). This complex solute is composed by surfactant in monomer state and surfactant in micellar estate. In the case of a 2-component system, the solvent will be the component 1 and the solute will be component 2.”

I appreciate the idea and work made by the authors. In my opinion, the article is certainly useful to the scientific community and deserves publication. The paper is well written and well structured. The title clearly describes the
contents of the paper. The abstract provides a concise and complete summary and the reference list is well chosen. The presentation is clear and the language is fluent and precise. 

I recommend this paper for publication. However, I do have a couple of comments that should be considered prior to publication:

1.line 49 sentence: "She also proposed a method calculating the enthalpy of 49
micelle formation ...." Not clear "who".

“She” was changed by Olofsson.

  1. I recommend deleting some obvious information about the ITC from the introduction.

I do not understand this remark

  1. Line 102, Line 226, Line 341: One style in symbols is recommended

When we refer a 2-component system, component 1 is the solvent and component 2 the solute. When we refer the micellar system, the subindex C16-TAB is the total solute complex. We include the following sentence around the old line 96:

By notation, in the work we will use the subindex C16-TAB to refer the total complex solute (surfactant). This complex solute is composed by surfactant in monomer state and surfactant in micellar estate. In the case of a 2-component system, the solvent will be the component 1 and the solute will be component 2.

  1. I would also mention that also time after which the prepared solution is used affects titration curve shapes. What I mean by that is that cmc values, as well as titration curve shapes, are different for experiments made after one hour one day, and one week.

Ok, then aging is important in these systems.

In my opinion, the cmc is not a point is rather a region.  

We agree.

Reviewer 3 Report

I read with a lot of interest the paper of Corea and cow.. The paper is the experimental evidence of what we know from the solution thermodynamics. I’d like to refer to the papers of Desnoyer and cow, one of the first people to measure the micellization enthalpy by calorimetric methods. His method is based on the measurement of the dilution heats of many surfactant solutions at different concentrations before and after the cmc. Then dilution data are used to build up the curves of the apparent and molar enthalpies of the solute vs. solute concentration, considering the infinite dilution as standard state. From these curves the values of the change in enthalpy upon micellization are obtained, by applying the phase separation model. With the use of the titration calorimeter, an instrumentation saving a huge quantity of time and work, this procedure has been lost and the parameter of interest in surfactant thermodynamics are directly obtained from the titration data. This is not wrong because we measure a difference and not an absolute value in the case of the change in enthalpy upon micellization, as data reported in table 3 shown, but the dilution curves must differ as a function of the stock solution concentration because the standard state now is the concentration of the stock solution itself.

I agree with the necessity of avoiding as much as possible the subjectivity in extracting data from the calorimetric titration curves and the proposed method is interesting and could work well with the “normal” surfactants, but I’m afraid it doesn’t work when, for instance, gemini surfactants or bile salts are studied.

It is not clear what the authors suggest, from an experimental point of view, to have reliable data, apart the use of theoretical models, probably based on mass action approach. I suggest to stress this aspect.

Some minor remark:

Move left eq. 3:

Line 126: remove shape

Line 135: remove be

Line 154: It is said “eight stock solution” but in table 1 only six are reported

Line 198: Panel A of figure 2, not 3

line 219:  Figure 1 and 2A, not 3A

line 226-227: as above

line 242: square brackets “by van Os [15]”

line 246: The used symbols in general are written in a different way in respect to those reported in equations and are quite difficult to understand

line 300: not there but these

line 345: add square brackets

line 407: explained not explain

line 425: add square brackets

line 461: erase depend

Author Response

Reviewer 3-report

References were checked and the following paragraphs were included:

Around old line 270: All bibliography data were obtained by isothermal titration calorimetry using an inflexion-point method of a break-point method [23,28,33,34,35-,36,37].

Arround old line 290: Panel B of Figure 8 compares the step height values obtained in this work and literature values calculated by isothermal titration calorimetry using similar methods [23,28,33,34,35,36,37,39] (Table 3 in Supplementary Material).

The following sentence was included arround the old line 96: “By notation, in the work we will use the subindex C16-TAB to refer the total complex solute (surfactant). This complex solute is composed by surfactant in monomer state and surfactant in micellar estate. In the case of a 2-component system, the solvent will be the component 1 and the solute will be component 2.”

You can find the new paragraphs in red in the new version of the manuscript. The anwers of your comments are listed bellow.

I read with a lot of interest the paper of Corea and cow.. The paper is the experimental evidence of what we know from the solution thermodynamics. I’d like to refer to the papers of Desnoyer and cow, one of the first people to measure the micellization enthalpy by calorimetric methods. His method is based on the measurement of the dilution heats of many surfactant solutions at different concentrations before and after the cmc. Then dilution data are used to build up the curves of the apparent and molar enthalpies of the solute vs. solute concentration, considering the infinite dilution as standard state. From these curves the values of the change in enthalpy upon micellization are obtained, by applying the phase separation model. With the use of the titration calorimeter, an instrumentation saving a huge quantity of time and work, this procedure has been lost and the parameter of interest in surfactant thermodynamics are directly obtained from the titration data. This is not wrong because we measure a difference and not an absolute value in the case of the change in enthalpy upon micellization, as data reported in table 3 shown, but the dilution curves must differ as a function of the stock solution concentration because the standard state now is the concentration of the stock solution itself.

I agree with the necessity of avoiding as much as possible the subjectivity in extracting data from the calorimetric titration curves and the proposed method is interesting and could work well with the “normal” surfactants, but I’m afraid it doesn’t work when, for instance, gemini surfactants or bile salts are studied.

In this work it is proposed that the exposed behavior could be general, including for non-interacting systems as consequence of the equation 3. At the moment, more experimental evidence is needed about this behavior with different systems.

It is not clear what the authors suggest, from an experimental point of view, to have reliable data, apart the use of theoretical models, probably based on mass action approach. I suggest to stress this aspect.

The following paragraph was included in the line 394 page 27 of the manuscript:

This discussion resembles, somehow, to the way used in ligand binding studies where the binding equilibrium constant and the binding enthalpy are calculated by different theoretical models. In the simplest case of the interaction of ligand and a macromolecule with a stoichiometry 1:1, studied by means of the Wiseman Isotherm [54], the characteristics of the titration curve depends dramatically on the Brandt´s parameter [54, 55] defined as:

where cM is the concentration of macromolecule and KBind is the binding equilibrium constant. Graphical methods are not used in this case because the shape and magnitude of the titration curve depends dramatically on the experimental design expressed in terms of the Brandt´s parameter. In addition to this, the value of this parameter affects the precision and reliability of the values of binding enthalpy and binding equilibrium constant obtained by fit of theoretical models and some authors have proposed different ranges of c in order of get the optimum value of this experimental parameter [55-56].

And the following paragraph was included in the line 397 of page 17:

As it was said, the titration curve approaches to the relative partial molar enthalpy of solute when the stock solution concentration increases. To use high stock solution concentrations could be a way to avoid systematic errors in the determinations of CMC and Dmich. The problem is that titration volumes would be very small and other systematic errors will appear due to the lack of control in the automatic titration system. From an operational point of view many times it is not possible to carry out a study similar to that of this work in order to determine the better value of the stock solution concentration. Based upon the fact that in general the shape and magnitude of the titration curve depend on the experimental design, the solution of this problem is to use theoretical model (as in the case of ligand binding). This is the most recommended procedure to determine the critical micelle concentration and the micellization enthalpy by isothermal titration calorimetry, in special for shorter alkyl chains for which the micellization transition is not so steep (cooperative). For this reason, theoretical models including the stock solution concentration effect [31] should be used to obtain the CMC and Dmich because if this effect is not considered [48], systematic errors could be present in the results.

Some minor remark:

Move left eq. 3:

I do not understand this remark.

Line 126: remove shape

Yes, in in line 116. The first word “Shape” was removed.

Line 135: remove be

“To be used” was changed by “to use them”.

Line 154: It is said “eight stock solution” but in table 1 only six are reported

“eight” was changed by “six”.

Line 198: Panel A of figure 2, not 3

“Figure 3” was changed by “Figure 2”.

line 219: Figure 1 and 2A, not 3A

“Figure 3” in line 194 was changed by “Figure 2”

line 226-227: as above

“Figure 4” was changed by “Figure 4”

line 242: square brackets “by van Os [15]”

Brachets were included in the reference

line 246: The used symbols in general are written in a different way in respect to those reported in equations and are quite difficult to understand

(Q/ns2)0 was changed by “(Q/nsC16-TAB)0

line 300: not there but these

I do not understand this remark

line 345: add square brackets

Brackets were included

line 407: explained not explain

“explain” was changed by “explained”.

line 425: add square brackets

Brackets were included

line 461: erase depend

“depend” was erased.
